# The Predictive Power of the Transplant Evaluation Rating Scale (TERS) for Psychosocial Outcomes in Living-Donor Kidney Transplant Recipients: A Two-Year Prospective Study

**DOI:** 10.3390/jcm13237076

**Published:** 2024-11-22

**Authors:** Ernst Peter Richter, Betty Schlegel, Hendrik Berth

**Affiliations:** Research Group Applied Medical Psychology and Medical Sociology, Division of Psychological and Social Medicine and Developmental Neurosciences, Faculty of Medicine, TU Dresden, 01307 Dresden, Germany; ernstpeter.richter@uniklinikum-dresden.de (E.P.R.); betty.schlegel1@mailbox.tu-dresden.de (B.S.)

**Keywords:** Transplant Evaluation Rating Scale (TERS), living-donor kidney transplant (LDKT), psychosocial outcomes, mental distress, physical complaints, perceived social support, predictive value, longitudinal study, transplant recipients, psychosocial evaluation

## Abstract

**Background/Objectives:** The Transplant Evaluation Rating Scale (TERS) assesses the psychosocial risk of transplant candidates; however, its predictive value for outcomes in living-donor kidney transplant (LDKT) recipients remains unclear. This study evaluated the predictive power of the TERS for psychosocial outcomes in LDKT recipients over two years post-transplant. **Methods:** In this prospective single-center cohort study, 107 LDKT recipients completed assessments pre-transplant (T0), 6 months post-transplant (T1), and 24 months post-transplant (T2). The outcomes measured were mental distress, physical complaints, and perceived social support. Linear mixed-effects models were used to examine the relationship between the pre-transplant TERS scores and outcomes over time. **Results:** Higher TERS scores predicted increased physical complaints (*p* < 0.001) and lower perceived social support (*p* = 0.035) at all time points. Additionally, higher TERS scores were associated with greater mental distress between T0 and T2 (*p* < 0.001). A hierarchical partitioning revealed that the TERS accounted for 11.9% of the variance in mental distress, 14.6% of that in physical complaints, and 6.0% of that in perceived social support. **Conclusions:** The pre-transplant psychosocial risk, as measured by the TERS, significantly predicted the psychosocial outcomes in the LDKT recipients over two years, with small-to-medium effect sizes. The TERS may serve as a valuable tool for identifying patients who could benefit from targeted psychosocial interventions to improve their long-term outcomes.

## 1. Introduction

Kidney transplantation (Tx) from living donors is a complex medical procedure that extends far beyond the surgical act itself, bringing about profound physical, psychological, and social changes for both patients and their families [1]. This procedure necessitates a holistic approach to patient care, wherein a preoperative psychosocial evaluation plays a crucial role. This evaluation aims to identify the potential psychopathological risks at an early stage that could significantly influence the long-term success of a transplantation and a patient’s overall well-being [2].

In Germany, pre-transplant psychosocial evaluations of transplant candidates are carried out in accordance with the standards set by the German Medical Association and the latest updates in the S3 guidelines, titled “Psychosocial Diagnostics and Treatment of Patients Before and After Organ Transplantation”. These guidelines emphasize the importance of a comprehensive and structured psychosocial assessment, ensuring that patients with insufficient psychosocial stability are either temporarily or permanently excluded from transplantation, or are classified as eligible but at an increased risk for adverse post-transplant outcomes [3,4]. A comprehensive preoperative psychosocial assessment requires a thorough evaluation of a patient’s personality profile, understanding of their illness, and the strength of their familial and social support systems. Psychosocial factors play a critical role in the success of transplantation, and if inadequately addressed, can significantly jeopardize the psychological stability of patients postsurgery. This is particularly true for living-donor kidney transplants, in which the recipient receives a kidney from a healthy individual with whom they have a close relationship, often a family member. This altruistic act has ethical implications that require special protective measures and careful psychosocial support for donors [5]. Kidney transplantation from living donors is governed by strict regulations in Germany to ensure the protection of both donors and recipients. Living donors must be adults, capable of giving informed consent, and medically suitable. Living donation is only allowed when no suitable postmortem organ is available, and the donor and recipient share a close personal relationship. While altruistic donation is rare, exceptions exist in cases of evident personal attachment [6].

A psychosocial evaluation aims to identify the relevant psychosocial factors that can impact postoperative well-being and health-related quality of life (HRQoL). Studies have shown that psychosocial stressors, such as preoperative anxiety, depression, and lack of social support, are significantly associated with poorer postoperative outcomes, including a lower HRQoL and higher rates of non-adherence. This is particularly important because adherence to immunosuppressive therapy and other medical recommendations post-transplantation is crucial for the long-term success of a transplantation and graft survival [7,8]. A psychosocial evaluation is conducted through interviews and often preceded by psychometric testing. In line with these guidelines, more than 20 aspects are assessed [3]. This typically culminates in a multi-page report. In response to the critical need for a structured psychosocial evaluation of transplant candidates, the Transplant Evaluation Rating Scale (TERS) was developed to systematically assess psychosocial risks. The primary advantage of the TERS is its ability to classify the psychosocial functioning of a patient across multiple domains and consolidate them into a single composite score. This score provides a clear, quantitative basis for assessing a patient’s psychosocial adaptation potential, which is particularly valuable in the selection of transplant candidates [9]. In various transplantation contexts, including heart, liver, lung, and stem cell transplants, the TERS has proven useful for identifying psychosocial risks and planning preoperative interventions that may improve postoperative outcomes [10,11,12]. A prospective study examining the prediction of the survival of patients on a heart transplant waiting list found that the TERS had a significant predictive power for survival rates. Higher TERS scores were significantly associated with increased mortality, underscoring the relevance of psychosocial evaluation for identifying high-risk patients. In a German study involving 85 patients awaiting liver transplantation that examined the predictive value of the TERS, the scale was found to have a significant discriminatory power, particularly among the patients with psychiatric diagnoses. The patients with psychiatric disorders had significantly higher TERS scores, indicating poorer psychosocial functioning. Higher TERS scores were also correlated with an increased likelihood of temporary exclusion from the transplant list [13]. Despite the evidence supporting the predictive value of the TERS in various transplant settings, research on its application to kidney transplantation, particularly for living-donor kidney transplantation recipients, remains scarce. Among the few studies that have explored the application of the TERS to kidney transplant recipients, the scale did not serve as a significant predictor of clinical outcomes, including renal function or the incidence of acute rejection episodes within the first year post-transplant [14]. However, these findings suggest that the TERS may potentially hold more predictive value for psychosocial outcomes, particularly in the context of living-donor kidney transplantation (LDKT), where the psychosocial dynamics between the donor and recipient are distinct.

Given the limited evidence on the predictive utility of the TERS for LDKT, this study sought to bridge this gap by evaluating the predictive value of TERS scores for psychosocial and physical outcomes after transplantation in living-donor kidney transplant recipients. Specifically, we aimed to assess whether the TERS can offer valuable insights into the key factors, such as psychological distress, physical complaints, and perceived social support, which are known to significantly influence long-term well-being and the health-related quality of life of transplant recipients [15,16]. Therefore, this study aimed to evaluate the predictive value of TERS scores concerning these psychosocial and physical outcomes in living-donor kidney transplant recipients over two years post-transplant.

## 2. Materials and Methods

### 2.1. Study Design and Participants

This prospective single-center cohort study was conducted at the Dresden Transplant Center, and 107 living-donor kidney transplant recipients were enrolled between 1 July 2011 and 31 December 2017. This study aimed to evaluate the predictive value of the Transplant Evaluation Rating Scale for the psychosocial outcomes in patients undergoing LDKT. This study included all patients who underwent a comprehensive psychosocial evaluation at the Department of Psychological and Social Medicine and Developmental Neurosciences at the University Hospital Carl Gustav Carus Dresden, followed by a successful LDKT. Patients who did not undergo an LDKT after a psychosocial evaluation or who received a kidney transplant from a deceased donor were excluded from this study (*N* = 59). This study’s protocol was reviewed and approved by the ethics committee of the medical faculty of TU Dresden (EK 186042015), ensuring that all the procedures adhered to the ethical standards for medical research. All the participants were fully informed of this study’s objectives and procedures and provided written informed consent before participation. All the participants underwent a comprehensive pre-transplantation psychosocial evaluation at the Department of Psychological and Social Medicine of the University Hospital, Carl Gustav Carus Dresden. These evaluations were conducted in conjunction with those of the living donors and involved both joint and separate interviews. The sessions were typically conducted during a single appointment and lasted approximately 2–3 h. An experienced medical psychologist or psychotherapist performed all the evaluations. The purpose of these sessions was to assess the various psychosocial factors that could influence health outcomes after transplantation.

Notably, all 107 patients were positively evaluated during the psychosocial assessment and subsequently recommended for living-donor transplantation. This positive evaluation indicated that the patients demonstrated sufficient psychosocial resilience and support to proceed with the transplantation process.

The pre-transplantation survey (T0) was administered approximately two weeks before the scheduled psychosocial evaluation, either via postal mail or email. During the in-person evaluation session, the completed questionnaires were thoroughly reviewed and discussed with the participants. On average, the psychosocial evaluation itself took place approximately six months before the transplantation surgery. Post-transplantation follow-up assessments were conducted at two specific time points. The first follow-up survey (T1) was conducted 6 months after the transplantation, and the second follow-up survey (T2) was conducted 24 months after the Tx. At both T1 and T2, questionnaires were sent to the living-donor kidney transplant (LDKT) recipients either by email or postal mail. At T0, six months before the transplantation, 107 living-donor kidney transplant recipients participated in this study. At the first follow-up (T1) at 6 months post-transplantation, data were collected from 94 of the 107 patients. At the second follow-up (T2), 24 months post-transplantation, the number of participants providing data had decreased to 74.

### 2.2. Measures

#### 2.2.1. Pre-Transplant Psychosocial Evaluation Interview

The psychosocial evaluation was designed to comprehensively assess a range of factors that could potentially influence the post-transplant outcomes of both the recipients and donors. This evaluation encompassed a broad spectrum of psychosocial aspects, including the identification of psychosocial stressors, such as preoperative anxiety, and other psychiatric disorders, such as depressive disorders, substance use, and fatigue. Moreover, the evaluation assessed the patients’ cognitive function and comprehension of their medical condition and the transplantation process. Furthermore, the evaluation examined their personal and social resources, focusing on their resilience, coping mechanisms, and the quality of the patients’ social support networks. It also evaluated health behaviors and lifestyle factors, including adherence to medical treatment and involvement with substance use. The patients’ motivation for undergoing transplantation and their decision-making processes were scrutinized to ensure that these decisions were well considered and made voluntarily.

#### 2.2.2. Outcome Measures

The Transplant Evaluation Rating Scale is an expert rating instrument designed to assess psychosocial functioning in transplant candidates across 10 domains [9]. Each domain is evaluated on a 3-point scale: good (1), moderate (2), or insufficient (3), depending on the severity of the symptoms present within each domain. These domains include a prior psychiatric history of DSM-III-R Axis I and II disorders, substance use, adherence, health behaviors, strength of family and social support networks, prior coping history, past coping strategies, affect quality, and mental status. For instance, a patient who consistently engaged in healthy behaviors, such as regular physical activity, nonsmoking, and a balanced diet, would be rated 1 on the health behavior domain. A detailed description of the complete rating system can be found in publication [9]. A patient who modified their health habits only after being diagnosed, for example, by quitting smoking after the diagnosis of terminal liver disease, would receive a score of 2. Meanwhile, a patient who continued with poor health behaviors, such as consuming a diet high in unhealthy fats, smoking, and excessive alcohol consumption, would be rated 3. After each domain was scored based on the clinical interview, the scores were weighted and the total score was calculated. The possible range of total scores was from 26.5 to 79.5, with lower scores indicating fewer psychosocial risk factors and, thus, better psychosocial functioning.

In 2021, the TERS was retrospectively applied to the pre-transplant psychosocial evaluation reports of 107 patients who had successfully undergone an LDKT transplantation by two independent raters from the Department of Psychosocial Medicine at the University Hospital Carl Gustav Carus, Dresden. This retrospective approach was chosen because the TERS was not routinely implemented in clinical practice during the time of data collection. By applying the TERS retrospectively, this study aimed to leverage comprehensive psychosocial evaluation reports that were already available, thus avoiding an additional burden on patients or interference with clinical workflows. Similar retrospective applications of the TERS have been employed in studies across various transplantation contexts, including kidney and bone marrow transplantation, to evaluate psychosocial risk factors using existing patient records [14,17].

Before conducting the TERS assessments, the raters underwent a structured training program that included a comprehensive review of the TERS manual and practice sessions, using anonymized patient reports. These sessions focused on the accurate application of the scale across the various domains. The raters were trained to ensure a consistent interpretation of the TERS criteria, particularly in complex cases. The raters independently conducted the TERS assessment, ensuring the objectivity and reliability of the ratings. Throughout the study period, recalibration sessions were conducted at regular intervals to maintain consistency in ratings. During these sessions, both raters independently re-evaluated patient reports from earlier in this study, and their new scores were compared to assess consistency and agreement between the raters. Any divergence in scoring was addressed through targeted feedback and discussion to realign their evaluations [18,19]. The inter-rater reliability, measured by Krippendorff’s alpha, was 0.82 for the total score, indicating a high degree of agreement between the raters [20]. This process provided a reliable assessment of the psychosocial risks associated with the transplant candidates, allowing for the identification of those who may require additional psychosocial support to improve their post-transplant outcomes.

Mental distress was evaluated using the Global Severity Index (GSI) from the Mini-SCL [21], the German adaptation of the Brief Symptom Inventory-18 (BSI-18). The BSI-18 is derived from the longer Symptom Checklist-90 (SCL-90), and retains the items that specifically target depression, anxiety, and somatization. The GSI provides an overall measure of psychological distress, calculated by totaling the scores of all 18 items, with higher scores indicating greater distress. This index captures the intensity of the symptoms experienced by a patient over the past seven days. In this study, the internal consistency of the GSI was strong, with a Cronbach’s alpha of 0.80 at T0, reflecting a high level of reliability. The Mini-SCL has demonstrated excellent internal consistency and strong validity in various populations, including kidney transplant patients [22,23].

Physical complaints were assessed using the total score (somatic symptom burden) from the Giessen Subjective Complaints List (GBB-24) [24]. The GBB-24 is a well-established and scientifically validated instrument used in German-speaking countries to evaluate psychosomatic physical complaints. The questionnaire comprises 24 items, each rated on a Likert scale ranging from “not at all” to “very much”, and addresses four key domains: cardiovascular complaints, gastrointestinal complaints, musculoskeletal complaints, and exhaustion. For this study, we focused on the total score and somatic symptom burden, which was derived from the sum of all 24 items. This score provides a comprehensive measure of the participants’ perceived physical health burden, reflecting the total intensity of the physical complaints reported. The internal consistency of the GBB-24 at T0 was excellent, with a Cronbach’s alpha of 0.89, indicating a high degree of reliability. The GBB-24 is widely used in psychosomatic medicine and psychotherapy, making it a reliable tool for capturing the extent of subjective physical complaints in diverse populations [25].

Perceived social support was assessed using the total score of the F-SozU-K-22, a short form of the widely used Social Support Questionnaire (F-SozU) [26]. The F-SozU-K-22 is composed of 22 items and is designed to efficiently measure perceived social support. It aggregates the various aspects of social support, such as emotional and practical support, as well as social integration, into a single global score that reflects the overall perceived social support. The internal consistency of the F-SozU-K-22 in this study was excellent, with a Cronbach’s alpha of 0.86 at T0, indicating a high degree of reliability. The F-SozU-K-22 is a validated tool that is particularly suitable for large-scale studies, in which a brief yet comprehensive measure of social support is required [27].

### 2.3. Statistical Analysis

Descriptive statistics were used to summarize the sample characteristics and TERS scores. The dichotomous and categorical variables were described as percentages (%), whereas the continuous variables were expressed as means (M). The primary analysis method applied was a univariable linear mixed-effects model (LMM) using the maximum likelihood estimation. This approach allowed for the assessment of the progression of different outcome measures over time and the influence of predictors. LMMs are particularly suitable for addressing random missing observations and the non-independence of repeated measurements within individuals, which are crucial for longitudinal studies. Unlike traditional statistical methods, such as ANOVA, which require complete data for all individuals, LMMs can handle missing random data efficiently using all the available information. This is achieved through the maximum likelihood estimation, which allows the model to make robust inferences even when some time points are missing for certain participants. This flexibility minimizes the bias and maintains the power of the analysis, ensuring more reliable estimates of the fixed and random effects over time. The models were constructed according to Cheng’s guidelines [28]. All the statistical analyses were performed in R Studio Version 4.3.2 (R Foundation, Boston, MA, USA) using the following packages: lme4 with maximum likelihood, lmerTest, nlme, MuMIn, and glmm.hp. The statistical significance level was set at *p* < 0.05.

#### 2.3.1. Unadjusted Analysis

We used a series of linear mixed-effects models for each outcome to analyze the trends of the three outcome variables over time. The hierarchical structure of the data was accounted for by nesting repeated observations (Level 1) within individuals (Level 2) across all the analyses [29]. The time was included as a fixed effect and the participants were treated as random effects to account for individual variability in the repeated measures. Given the potential nonlinear relationship between the time and outcomes, time was treated as a categorical variable. The period between the pre- and post-Tx may have witnessed a shift in the patients’ medical and psychosocial states. Treating time as categorical variable allowed for a clearer distinction between the pre-transplant (T0) and immediate postoperative phases (T1), as well as the longer-term follow-up period (T2) [30]. The unadjusted analysis provided a broad overview of the group trends over time, averaging the outcomes across the entire sample. This analysis enabled a comparison of the estimates before and after a covariate adjustment, providing a more nuanced understanding of the data.

#### 2.3.2. Adjusted Analysis

To assess whether the Transplant Evaluation Rating Scale could predict different psychosocial outcome trajectories over time, we conducted a series of adjusted linear mixed-effects models. These models were designed to explore whether the TERS scores predicted changes in the outcomes during the post-transplant period. Time was treated as a categorical variable, allowing us to examine the predictive effect of the TERS at specific time points post-transplantation. The TERS score, a continuous variable, was grand-median-centered to account for its right-skewed distribution. This centering ensured that the intercept represented the baseline outcome for a patient with a median TERS score [31]. In these models, the fixed effects included the time, TERS, and interaction term (TERS × time). The fixed effects captured the average relationship between these predictors and the psychosocial outcomes, revealing the overall trends and group-level patterns over time. Specifically, the interaction term allowed us to examine whether the relationship between the TERS score and the outcomes varied at different time points. The random effects were modeled on the intercept, allowing us to account for individual differences in the baseline outcomes. This random intercept helped capture the variability among the patients, acknowledging that each individual’s starting point or overall trajectory might differ from the average trend identified by the fixed effects.

#### 2.3.3. Variance Explanation and Hierarchical Partitioning Derived from Adjusted Analysis

To estimate the effect size in the LMMs, we applied the pseudo-R^2^ method suggested by Nakagawa and Schielzeth [32], and calculated the explained variance using the r.squaredGLMM function in the MuMIn package [33]. The marginal R^2^ (mR^2^) represents the proportion of the total variance explained by the fixed effects only. By contrast, the conditional R^2^ (cR^2^) reflects the total variance explained by both the fixed and random effects. To interpret the effect sizes, the mR^2^ of terms with values around 0.02 typically indicates a small effect, values around 0.13 represent a medium effect, and values of 0.26 or higher suggest a large effect [34]. Although LMMs highlight group-level differences and individual trajectories, they do not reveal the unique contribution of each predictor to the total variance. Hierarchical partitioning addresses this problem by decomposing the variance and identifying the independent contribution of each predictor. This method quantifies how much variance each predictor explains independently, offering deeper insight into which factors drive the marginal variance (mR^2^). We used the glmm.hp package for hierarchical partitioning [35,36], following Chevan and Sutherland’s [37] approach, to rank the predictors based on their explanatory power. Although this method is frequently employed in ecological studies [38,39], hierarchical partitioning remains relatively novel in the health sciences, particularly when applied to hierarchical models such as LMMs [40].

## 3. Results

### 3.1. Descriptive Statistics

A total of 107 living-donor kidney transplant recipients participated in this study at the baseline (T0). The average age of the participants was 48.0 years, and 37.4% were female. Additionally, 81.3% of the participants were living with a steady partner, and 76.6% had children. All the participants were of Caucasian ethnicity. The detailed sociodemographic characteristics are shown in Table 1. The TERS rating was conducted retrospectively for all 107 recipients. The TERS scores ranged from 26.5 to 51.0; the mean score was 30.0, the median was 29.0, and the standard deviation was 4.6. The interquartile range (IQR) of the TERS scores was 5, indicating that the middle 50% of the scores fell within a relatively narrow range. The distribution of the TERS scores exhibited positive skewness (2.1), indicating that the data were right-skewed, with a longer tail towards higher scores. Additionally, the kurtosis value of 5.2 suggests that the distribution was leptokurtic, meaning it had a sharper peak and heavier tails than a normal distribution. This indicates the presence of extremely high values that pulled the distribution to the right [18].

### 3.2. Unadjusted Analysis: General Trend of Psychosocial Outcomes over Time

Figure 1 depicts the mean trajectories for mental distress, physical complaints, and perceived social support at 6 months pre-transplant, 6 months post-transplant, and 24 months post-transplant in the LDKT recipients.

Table 2 presents the *p*-values from the unadjusted linear mixed models across the time points, along with the explained variance. For mental distress, the LMM results showed no significant change from T0 to T1 (*p* = 0.574), whereas no significant increase was observed from T0 to T2 (*p* = 0.064). For physical complaints, there was a statistically significant reduction from T0 to T1 (*p* < 0.001), followed by a return to pre-transplant levels, with no significant difference between the pre-transplant levels (*p* = 0.645). In terms of the perceived social support, no significant difference was observed between T0 and T1 (*p* = 0.175); however, a significant reduction was observed between T0 and T2 (*p* = 0.010).

### 3.3. Adjusted Analysis: The Predictive Value of the TERS Rating for the Psychosocial Outcomes Post-Transplant

The adjusted analysis examined how the pre-transplant TERS rating predicted changes in mental distress, physical complaints, and perceived social support for up to 24 months post-transplantation. Table 3 summarizes the results of the adjusted analysis.

For mental distress, no significant change was observed from T0 to T1 (*p* = 0.457) or from T0 to T2 (*p* = 0.820) in the patients with median TERS scores, indicating stable mental distress levels over time. The TERS score was not significantly associated with the baseline mental distress at T0 (*p* = 0.165), indicating that the TERS did not predict pre-transplant mental distress. However, the interaction between Time_T0T2 and the TERS was significant from T0 to T2 (*p* < 0.001), suggesting that patients with a higher TERS experienced a significant increase in mental distress between 6 and 24 months post-transplantation.

For physical complaints, a significant reduction was observed from T0 to T1 (*p* < 0.001), with no significant difference between T0 and T2 (*p* = 0.347), indicating a temporary improvement post-transplantation that returned to pre-transplant levels by 24 months. The TERS was a significant predictor across all time points (*p* < 0.001), indicating that a higher psychosocial risk consistently predicted greater physical complaints, although the interaction between the time and the TERS was not significant (*p* = 0.210).

For perceived social support, no significant change was noted from T0 to T1 (*p* = 0.378), but a significant decrease was observed from T0 to T2 (*p* = 0.002). The patients with higher TERS scores consistently reported lower perceived social support at all time points (*p* = 0.035). The interaction between the time and the TERS was not significant (*p* = 0.228), indicating the consistent relationship between the TERS and the perceived social support over time.

### 3.4. Total Variance Explained and Decomposition of Explained Variance with Hierarchical Partitioning

To assess the predictive value of the Transplant Evaluation Rating Scale (TERS) for psychosocial outcomes post-transplant, we compared the unadjusted and adjusted linear mixed-effects models (LMMs). By comparing the marginal R^2^ values, which represent the variance explained by the fixed effects, we measured the additional contribution of the TERS to predicting each outcome. The inclusion of the TERS in the adjusted models increased the explained variance by 11.9% for mental distress (small effect), 14.6% for physical complaints (medium effect), and 6.0% for perceived social support (small effect). 

These findings suggest that while the TERS significantly contributes to predicting all psychosocial outcomes, its predictive utility is most pronounced for physical complaints.

A hierarchical partitioning analysis allowed us to understand which factors had the greatest impact on the patients’ post-transplant experiences. For mental distress, the TERS explained 39.6% of the marginal variance, making it a significant predictor of the post-transplant mental burden. This suggests that the patients who exhibited a higher psychosocial risk (as indicated by the TERS) experienced greater levels of mental distress across all the time points. Notably, the interaction between the TERS and time (T0–T2) accounted for 50.6% of the marginal variance, indicating that the predictive ability of the TERS for mental distress became more pronounced between 6 and 24 months post-transplant. This means that, while the psychosocial risk was already a concern at the baseline, its influence on mental distress tended to grow over time, particularly for those at higher risk. For physical complaints, the TERS explained 70.3% of the marginal variance, underscoring the substantial role of psychosocial risk in determining patients’ physical symptom burdens, both before and after transplantation. Patients with higher TERS scores consistently reported more physical complaints at all the time points. Furthermore, the time (T0–T1) explained 20.1% of the variance, indicating that physical complaints improved significantly six months post-transplant. Despite this initial improvement, the effect of the psychosocial risk remained significant, with patients reporting a higher burden of physical symptoms over time. For perceived social support, the TERS explained 54.9% of the marginal variance, showing that patients with higher psychosocial risk consistently felt less supported, both before and after transplantation. This underscores the importance of the psychosocial risk in shaping patients’ perceptions of their social support during the post-transplant period. Additionally, the time (T0–T2) accounted for 24.2% of the variance, highlighting a significant decline in perceived social support by 24 months post-transplantation. This decline was especially pronounced for those with higher TERS scores, suggesting that patients with a greater psychosocial risk are more likely to feel isolated or unsupported as time progresses, which could negatively impact their overall recovery and well-being. Figure 2 illustrates the relative importance of each predictor according to the hierarchical partitioning.

## 4. Discussion

To our knowledge, this study is the first to evaluate the predictive value of the Transplant Evaluation Rating Scale for psychosocial outcomes in living-donor kidney transplant recipients over a two-year post-transplantation period. Our findings offer a comprehensive understanding of the natural progression of psychosocial outcomes over time and demonstrate how the pre-transplant psychosocial risk, as measured by the TERS, influences these outcomes in patients for up to two years after transplantation. We observed in the LDKT recipients a general trend of a significant reduction in physical complaints from pre-transplantation to 6 months post-transplantation, followed by a return to the baseline levels by 24 months. Additionally, the perceived social support significantly decreased from 6 to 24 months, whereas mental distress remained relatively stable throughout the study period. These findings indicate that while patients may experience a short-term physical recovery, they may also feel less socially supported two years after a transplantation. Furthermore, our adjusted analyses revealed important insights into how the pre-transplant psychosocial risk, as measured by the TERS, influences post-transplant outcomes. Higher TERS scores were consistently associated with more physical complaints and lower levels of perceived social support across all time points, including pre-transplantation. This indicates that patients with a higher psychosocial risk had already experienced worse physical health and felt less socially supported before the transplant, and these differences persisted over the two-year follow-up period. In contrast, the relationship between the TERS scores and mental distress followed a different pattern. There was no significant association between the TERS scores and mental distress at T0, indicating that patients with varying levels of psychosocial risk did not differ in their pre-transplant mental distress. However, the interaction between the TERS score and time was significant between T1 and T2. This means that a higher pre-transplant psychosocial risk was linked to an increase in mental distress between six months and two years post-transplantation. Thus, the influence of the psychosocial risk on mental distress intensified over time. Notably, our hierarchical partitioning analysis revealed that the TERS accounted for a substantial proportion of the explained variance for all outcomes, with small-to-medium effect sizes in terms of the marginal explained variance. Specifically, when considering both the TERS and its interaction with time, 93.5% of the variance was in mental distress, 77.8% was in physical complaints, and 64% was in perceived social support outcomes, relative to the variance explained by the time effect alone. This highlights the significant predictive power of pre-transplant psychosocial risk factors for the long-term progression of these outcomes, far surpassing the explanatory power of time, which primarily captures natural recovery patterns and general postsurgical health improvements.

Our findings align with those of several key studies that have examined psychosocial outcomes in LDKT recipients. In our unadjusted analysis, we observed no significant changes in mental distress or perceived social support between pre-Tx and six months post-Tx. This is consistent with the PI-KT study [41], which reported stable levels of mental distress and social support after an LDKT. Regarding long-term mental health, our adjusted analysis showed that a higher psychosocial risk, as measured by the TERS, was linked to increased mental distress between 6 and 24 months post-transplant. This finding is consistent with the results of Goetzmann et al. [42], who examined patients who underwent lung, liver, and bone marrow transplantations. Using a cluster analysis, the authors identified two distinct patient clusters: one with generally good mental and physical health and another with significantly poorer outcomes. Importantly, their study revealed that while both clusters exhibited similar mental health pre-transplant, by six months post-transplant, the cluster with poorer outcomes began to experience a notable decline in mental health, which persisted throughout the two-year follow-up. This pattern corroborates our finding that the pre-transplant psychosocial, while not predictive of initial mental distress, emerged as a critical determinant of mental health deterioration during the long-term follow-up period. Conversely, Dobbels et al. [43] reported rising depression rates among kidney transplant recipients, increasing from 5.1% at one year to 9.1% at three years post-Tx. The patients who developed depression had a more than two-fold increased risk of graft failure and death with a functioning graft than those without depression. While our study did not observe a generally significant deterioration in mental distress across the entire sample, we found that the patients with higher TERS scores experienced a notable increase in mental distress between 6 and 24 months post-transplantation. This suggests that similar to the findings above, mental health issues may emerge or worsen over time in patients with higher psychosocial risk. Previous studies, including those by Broers et al. and Mitsui et al., have shown significant improvements in the physical HRQol within six months after an LDKT [44,45]. Similarly, our unadjusted analysis found a significant reduction in physical complaints during this period, highlighting the benefits of early physical recovery following a kidney Tx. Specifically, during the first few months after a transplantation, the patients experienced a substantial benefit from the transplant procedure and no need for dialysis. Our study also found that higher TERS scores were associated with lower perceived social support consistently over 24 months. The decline in perceived social support may be because post-transplant recipients no longer need the intensive support they required during dialysis or while waiting for a transplant. As their physical health improves, family and social networks may reduce their level of involvement [46]. Prihodová et al. emphasized the importance of social support after transplantation and found that higher perceived family support was linked to better adherence to immunosuppressive medication and a reduced risk of graft loss and mortality over 12 years [47].

Few studies have explored the use of the TERS in kidney transplantation. Previous research has found no significant correlation between the TERS scores and medical outcomes, such as estimated glomerular filtration rate decline or acute rejection within the first year post-transplantation [14]. However, our findings suggest that higher TERS scores predict increased mental distress and decreased perceived social support over time, suggesting that the TERS may be more effective at forecasting psychosocial than medical outcomes in LDKT recipients. In other organ transplants, higher TERS scores have been associated with poorer psychosocial health. Goetzmann et al. found that higher TERS scores were negatively associated with perceived social support before transplantation and predicted a greater need for psychosocial counseling at 12 months in lung, liver, and bone marrow transplant recipients [48]. Similarly, Brewer et al. reported that higher pre-transplant TERS scores predicted a poorer quality of life and higher anxiety in stem cell transplant patients [49].

To the best of our knowledge, this investigation represents the first assessment of the Transplant Evaluation Rating Scale’s predictive validity for psychosocial outcomes in LDKT recipients. This is a key strength, as it addresses a gap in the existing literature by focusing on the unique psychosocial dynamics of LDKT recipients, where the donor–recipient relationship may often play a critical role in post-transplant recovery and long-term well-being. The extended 2 year follow-up period allows for a more comprehensive understanding of how the psychosocial risk factors identified pre-transplant impact patients over the long term, beyond the immediate post-transplant recovery phase. The application of advanced statistical techniques, particularly linear mixed-effects models, further strengthens this study. LMMs allow for the analysis of individual variability alongside group-level trends, accommodating the repeated-measures design and hierarchical nature of the data [29]. By treating the TERS as a continuous variable, we are able to capture the full range of psychosocial risk, offering a more precise assessment than previous studies that categorized patients into predefined groups (e.g., high or low risk). The unadjusted analyses provide an overview of the general trends for how psychosocial outcomes evolve post-transplant, whereas the adjusted models account for the unique contribution of the TERS to these outcomes.

Our study has several limitations. First, the single-center design and modest sample size limits the generalizability of our results. Additionally, since all the participants were Caucasian, the findings may not be applicable to other ethnic groups with different psychosocial characteristics. Another potential limitation of this study is the presence of unmeasured confounding variables, such as socioeconomic status or comorbidities, which were not included in the analysis. However, this exclusion was intentional, as the primary goal was to focus on the predictive value of the TERS. Another limitation is that LDKT recipients generally represent a healthier and more selectively chosen population than deceased-donor kidney transplant recipients. LDKT candidates often undergo transplantation earlier in their disease course, sometimes even before dialysis initiation, and must complete thorough physical and medical evaluations before the psychosocial assessment. This could limit the comparability of our findings to those of other transplant populations. Additionally, a key limitation of our study is the retrospective nature of the TERS rating. While experienced raters used detailed psychosocial reports and medical records to conduct the ratings, a prospective approach could provide richer and more immediate data. Performing TERS assessments during the psychosocial evaluation itself would allow for more nuanced information via direct patient interaction, likely enhancing its predictive accuracy and yielding stronger effect sizes. Despite this, we took significant steps to ensure the reliability and objectivity of the retrospective assessments by offering structured rater training, independent evaluations, and regular recalibration sessions throughout this study to maintain consistency. Importantly, prior studies have also employed retrospective TERS assessments and demonstrated their validity for predicting medical and psychosocial outcomes. For example, Dieplinger et al. [14] successfully applied a retrospective TERS scoring to explore correlations with post-transplant outcomes in living-donor kidney transplant recipients, highlighting the feasibility of this approach. This suggests that a retrospective TERS analysis can still yield meaningful insights, though prospective validation remains a priority for future research.

Future studies should focus on prospective designs with longer follow-up periods, such as five years or more, to capture the long-term predictive value of the TERS for both psychosocial and medical outcomes. Incorporating additional outcome measures, such as health-related quality of life, could provide a more comprehensive understanding of post-transplant well-being. Moreover, comparing the TERS with other psychosocial assessment tools, such as the Stanford Integrated Psychosocial Assessment for Transplantation (SIPAT) [50], could help to determine which instruments offer the most robust predictive power for both psychosocial and medical outcomes. Further research should explore the role of the TERS in diverse populations, as psychosocial factors and transplant outcomes may vary across cultural contexts. Additionally, future studies should investigate the predictive value of the TERS in larger, higher-risk cohorts, including patients with more complex psychosocial profiles or those undergoing deceased donor kidney transplantations. This would help to refine the identification of high-risk patients who may benefit from targeted interventions.

## 5. Conclusions

This two-year prospective cohort study highlights the significant predictive power of the Transplant Evaluation Rating Scale for psychosocial outcomes in living-donor kidney transplant recipients. The TERS effectively predicted increased physical complaints, lower perceived social support, and heightened mental distress over the follow-up period, with small-to-medium effect sizes for each outcome. Moreover, our hierarchical partitioning analysis revealed that the TERS and its interaction with time explained 93.5% of the marginal variance in mental distress, 77.8% of that in physical complaints, and 64% of that in perceived social support. These findings underscore the substantial importance of the TERS-assessed pre-transplant psychosocial risk factors for predicting long-term patient outcomes, far exceeding the explanatory power of time alone, which largely captures the natural post-transplant recovery trajectory.

Our results suggest that preoperative psychosocial assessments using the TERS can provide essential insights into which patients may benefit from targeted interventions aimed at improving their psychosocial health. Identifying high-risk patients early in the transplant process allows for better-tailored support, which may improve the long-term recovery and quality of life in LDKT recipients. This study adds to the growing body of literature supporting the critical role of psychosocial evaluation in enhancing transplant outcomes.

## Figures and Tables

**Figure 1 jcm-13-07076-f001:**
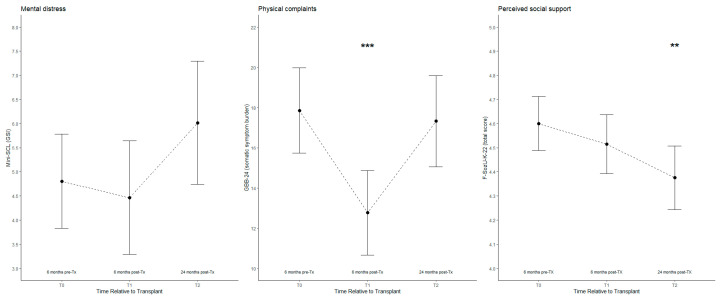
Trajectories for psychosocial outcomes from pre-transplant up to 2 years post-transplant for LDKT recipients (*N* = 107). Note: means of mental distress (Mini-SCL (GSI)), physical complaints (measured using GBB-24 (somatic symptom burden)), and perceived social support (F-SozU-K-22 (total score)). Error bars represent 95% confidence intervals. Significant changes from fixed effects of unadjusted LMMs are highlighted with asterisks (*** *p* < 0.001, ** *p* < 0.01; T0 as reference).

**Figure 2 jcm-13-07076-f002:**
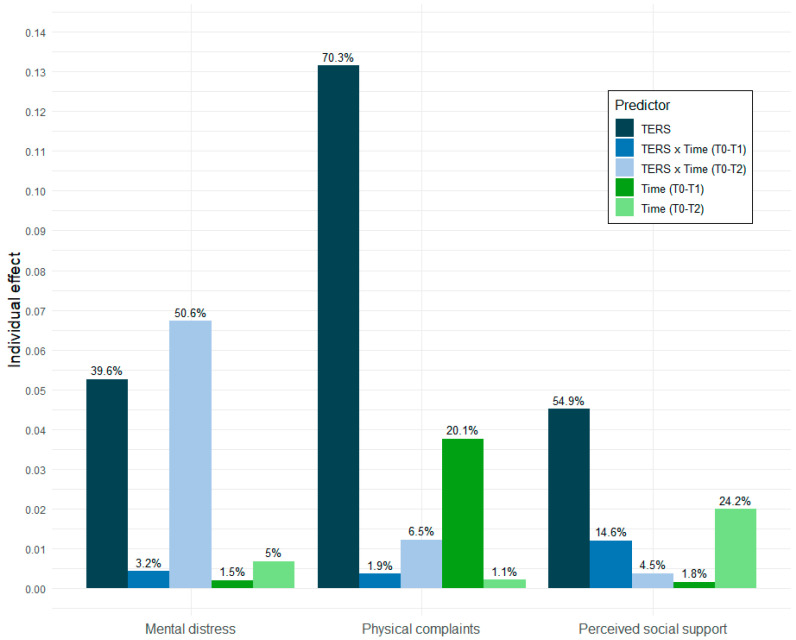
Relative importance of outcome variable predictors based on hierarchical partitioning analysis.

**Table 1 jcm-13-07076-t001:** Sociodemographic characteristics of sample at T0 (*N* = 107).

Gender (*n* (%))	
-Male	67 (62.6)
-Female	40 (37.4)
Age (mean)	48.0
Age (standard deviation)	11.2
Age (range)	22–73
Has children (*n* (%))	82 (76.6)
-Number of children (mean)	1.7
Family status (*n* (%))	
-Married	72 (67.4)
-Steady partner	15 (14.0)
-Single	10 (9.3)
-Divorced/separated	10 (9.3)
Level of education (*n* (%))	
-University	8 (7.5)
-Technical college	17 (15.9)
-Vocational training	80 (74.7)
-No vocational training	2 (1.9)
Current employment status (*n* (%))	
-Employed—full time	86 (40.2)
-Employed—part time	6 (5.7)
Not employed	
-In education/training	1 (0.9)
-Unemployed	7 (6.5)
-Disability pension	30 (28.0)
-Retired	8 (7.5)
-Other	12 (11.2)

**Table 2 jcm-13-07076-t002:** *p*-values and explained variance (*R*^2^) for unadjusted LMMs across time points (*N* = 107).

Dependent Variable	*p*-Value (T0-T1)	*p*-Value (T0-T2)	*mR* ^2^	*cR* ^2^
Mental distress	0.574	0.064	0.014	0.339
Physical complaints	<0.001 ***	0.645	0.041	0.578
Perceived social support	0.175	0.010 **	0.023	0.472

Note. T0 = 6 months pre-transplant; T1 = 6 months post-transplant; T2 = 24 months post-transplant; *mR*^2^ = marginal *R*^2^; *cR*^2^ = conditional R^2^; mental distress (Mini-SCL (GSI)); physical complaints (measured using GBB-24 (somatic symptom burden)); and perceived social support (F-SozU-K-22 (total score)). Significant changes from fixed effects of unadjusted LMMs are highlighted with asterisks (*** *p* < 0.001, ** *p* < 0.01; T0 as reference).

**Table 3 jcm-13-07076-t003:** Fixed effects, mR^2^, and cR^2^ for the adjusted linear mixed model (*N* = 107).

Dependent Variable		Estimates	*SE*	*t*	*p*
Mental distress	Intercept	4.5177	0.510	8.855	<0.001 ***
	Time_T0T1	−0.464	0.623	−0.745	0.457
	Time_T0T2	0.154	0.679	0.228	0.820
	TERS_mc	0.143	0.103	1.391	0.165
	Time_T0T1 x TERS_mc	0.082	0.128	0.644	0.520
	Time_T0T2 x TERS_mc	0.534	0.135	3.942	<0.001 ***
	Marginal R^2^	0.133			
	Conditional R^2^	0.395			
Physical complaints	Intercept	16.171	1.095	14.775	<0.001 ***
	Time_T0T1	−5.170	1.148	−4.503	<0.001 ***
	Time_T0T2	−1.178	1.249	−0.943	0.347
	TERS_mc	0.840	0.221	3.806	<0.001 ***
	Time_T0T1 x TERS_mc	0.059	0.235	0.250	0.803
	Time_T0T2 x TERS_mc	0.315	0.250	1.259	0.210
	Marginal R^2^	0.187			
	Conditional R^2^	0.587			
Perceived social support	Intercept	4.651	0.060	77.139	<0.001 ***
	Time_T0T1	−0.059	0.067	−0.883	0.378
	Time_T0T2	−0.232	0.073	−3.185	0.002 **
	TERS_mc	−0.026	0.012	−2.120	0.035 *
	Time_T0T1 x TERS_mc	−0.017	0.014	−1.210	0.228
	Time_T0T2 x TERS_mc	0.002	0.015	0.163	0.871
	Marginal R^2^	0.083			
	Conditional R^2^	0.475			

Note. Mental distress measured using the Mini-SCL (GSI); physical complaints assessed with the GBB-24 (somatic symptom burden); and perceived social support evaluated with the F-SozU-K-22 (total score). Time_T0T1 represents the change from T0 (pre-transplant) to T1 (6 months post-transplant), and Time_T0T2 represents the change from T0 to T2 (24 months post-transplant). TERS_mc refers to the grand-median-centered TERS score at T0. *mR*^2^ = marginal *R*^2^ (variance explained by fixed effects); *cR*^2^ = conditional *R*^2^ (variance explained by both fixed and random effects). Significance levels: * *p* < 0.05, ** *p* < 0.01, and *** *p* < 0.001.

## Data Availability

Data are available from the corresponding author upon request.

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
