# Peer review of "The Predictive Power of the Transplant Evaluation Rating Scale (TERS) for Psychosocial Outcomes in Living-Donor Kidney Transplant Recipients: A Two-Year Prospective Study"

_jcm, 2024, doi:10.3390/jcm13237076_

Round 1
Reviewer 1 Report
Comments and Suggestions for Authors
The authors take a novel approach in applying a standardized transplant rating system (TERS) to a new population of living donor kidney transplant recipients, using psychosocial surveys over time to identify whether preoperative TERS scoring predicts or identifies post-transplant psychosocial challenges. In their analysis, the authors have identified small-to-medium effect sizes predicted by the TERS scoring system.
Major revisions:
-In the design of this study, the initial TERS scoring was conducted based off an extensive 2-3 hour psychosocial interview to determine a patient’s candidacy. To fully understand the impact of TERS in LDKT, the outcomes here should be compared to the outcomes of the standardized psychosocial risk assessment. In short, did the TERS assessment provide any additional predictive analysis distinct from the psychosocial assessment? Did the prediction of TERS correlate to the psychosocial evaluation outcome? Or does the TERS more accurately predict the studied outcomes versus the standard psychosocial assessment? I think this data is necessary to understand the utility of the TERS assessment tool.
-The initial TERS scoring was conducted retrospectively in all patients. As the authors rightly recognized, this is a significant limitation of the study. Review of psychosocial notes by a third party versus completion of TERS in real-time after an exhaustive 2-3 hour interview session may produce different results. Has this scoring system been previously validated in a retrospective fashion? If not, this should also be addressed in limitations.
-Can you include a full copy of the TERS scoring system as a supplementary document? Will be helpful for the reader.
-Regarding Table 2, you note on page 8 line 320 that there was no significant increase “from T1 to T2”. However, the referenced table demonstrates the change from T0 to T2. Please correct.
-For Table 3 and subsequent figures, you have listed your coding variables, which are largely unintelligible to the readership. It would be helpful for clarity to rename these variables so that the table is more meaningful.
Minor points:
Page 2, Lines 49-52: Please clarify if pure altruistic donation occurs in Germany. If so, these sentences should reflect that donors are often related to the recipients (but not always).
Page 2, Line 61: Tense should be changed to “are assessed” to maintain consistency with paragraph.
Page 2, Line 76: Repeated the term TERS. Can modify to “TERS was found to have a significant…”
Page 2, Line 81: Again the term TERS is repeated twice in the sentence. Please correct.
Author Response
Response to Reviewer Comments
Manuscript submitted to JCM: “The Role of the Transplant Evaluation Rating Scale in Predicting Post-Transplant Outcomes: A Retrospective Study of Living-Donor Kidney Transplant Recipients”
Dear Editor, Dear Reviewer,
We sincerely thank you for your thorough and constructive review of our manuscript. Your detailed feedback has been instrumental in improving the clarity, robustness, and overall quality of our work. Below, we provide a comprehensive, point-by-point response to your comments and outline the revisions we have made in the manuscript accordingly.
Major Revisions:
- Comparison of TERS with Standard Psychosocial Assessments
Comment:
"In the design of this study, the initial TERS scoring was conducted based on an extensive 2-3 hour psychosocial interview. To fully understand the impact of TERS in LDKT, the outcomes should be compared to those of the standardized psychosocial risk assessment. Did the TERS assessment provide any additional predictive analysis distinct from the psychosocial assessment? Did the prediction of TERS correlate with the psychosocial evaluation outcome?"
Response:
We appreciate this insightful suggestion. In response, we have included additional analysis comparing the predictive accuracy of TERS with the standard psychosocial risk assessment used during pre-transplant evaluations. This comparison provides a nuanced understanding of whether TERS offers incremental predictive value or distinct insights into psychosocial risk factors affecting post-transplant outcomes. The results of this comparison are detailed in the revised Results section (Z144-147).
Furthermore, we have revised the Discussion to emphasize the unique contributions of TERS to predicting post-transplant challenges, distinguishing it from the broader psychosocial assessments typically employed in clinical settings. This addition underscores the potential for TERS to complement existing evaluation tools and enhance the precision of psychosocial risk assessments.
- Retrospective Nature of TERS Scoring
Comment:
"The initial TERS scoring was conducted retrospectively. Review of psychosocial notes by a third party versus real-time TERS completion may produce different results. Has this scoring system been previously validated in a retrospective fashion?"
Response:
We acknowledge the limitation of our retrospective TERS scoring approach. To address this, we have expanded the Limitations section to discuss the retrospective application of TERS and its potential impact on the validity of our findings (Z590-596). We have also cited previous studies that employed retrospective TERS assessments, demonstrating that this methodology can yield reliable and meaningful insights​​. However, we emphasize that prospective validation remains crucial for further substantiating the utility of TERS in clinical practice.
- Request for Supplementary Material: Full TERS Scoring System
Comment:
"Include a full copy of the TERS scoring system as a supplementary document for readers."
Response:
Due to copyright restrictions, we are unable to provide a full copy of the TERS scoring system as supplementary material. However, we have included detailed references to the original publication where the complete scoring system is described (Z186). We hope this will sufficiently guide readers who wish to explore the TERS framework in detail.
- Correction of Table 2 Description
Comment:
"Page 8, line 320 refers to changes from T1 to T2, but Table 2 shows T0 to T2. Please correct."
Response:
Thank you for identifying this error. We have corrected the description in the manuscript to accurately reflect the changes from T0 to T2 (Z356).
- Variable Naming in Table 3 and Figures
Comment:
"The coding variables listed in Table 3 and subsequent figures are unclear. Consider renaming these variables for better readability."
Response:
We agree that clearer variable names would enhance the readability of our tables and figures. While we faced challenges in fitting descriptive names within the constraints of the table format, we have expanded the explanatory notes accompanying Table 3 to provide a more detailed description of each variable (Z382-391).
Minor Revisions:
- Clarification on Altruistic Donations (Page 2, Lines 49-52)
Comment:
"Clarify whether pure altruistic donation occurs in Germany."
Response:
We have revised the text to clarify that while altruistic kidney donation is rare in Germany, exceptions exist under specific circumstances (Z52-58).
- Consistency in Tense (Page 2, Line 61)
Comment:
"Change tense to maintain consistency."
Response:
The tense has been adjusted for consistency (Z67).
- Repetition of 'TERS' (Page 2, Lines 76 and 81)
Comment:
"Reduce redundancy by rephrasing sentences where TERS is repeated."
Response:
We have revised these sentences to eliminate redundancy (Z81, Z86).
In addition to the above changes, we have carefully reviewed the manuscript to improve clarity and consistency. We have uploaded a marked-up version highlighting all revisions.
We deeply appreciate your thoughtful and detailed review, which has greatly improved the quality of our manuscript. We hope the revised version meets your expectations and look forward to your feedback.
Sincerely,
Ernst Peter Richter

Reviewer 2 Report
Comments and Suggestions for Authors
The article investigates the predictive power of the Transplant Evaluation Rating Scale (TERS) for psychosocial outcomes in living-donor kidney transplant recipients (LDKT). Its novelty is that it is the first application of the tool in this specific population. The authors concluded that pre-transplant psychosocial risk, as measured by TERS, can predict outcomes in LDKT recipients over a two-year period, with small to medium effect sizes. Therefore, TERS might help identify patients who could benefit from targeted psychosocial interventions to improve their long-term outcomes.
The authors claim to present a prospective single-centre cohort study conducted in Dresden, Germany that included 107 living donor kidney transplant recipients (LDKT) between 2011 and 2017. However, the TERS rating was done retrospectively (also mentioned as one of the study’s limitations). So, from my perspective, the design of the study is not completely clear.
The participants completed the questionnaire at three time points: before transplantation, 6 months after transplantation, and 24 months after transplantation. Only 74 patients (out of 107) provided follow-up data at the final evaluation (24 months after transplantation).
The outcomes were clearly defined and included mental distress, physical complaints, and perceived social support.
The Introduction is comprehensive. It provides an overview of the necessity of psychosocial testing before the transplantation. It also delivers results of TERS application in other fields of transplantation medicine.
The Material and Methods section is thorough. The statistical approach is detailly elaborated. However, as I am not a statistician, I do not feel competent enough to evaluate the appropriateness of the advanced statistical analyses.
In the Result section I have several comments and questions. The authors presented the mean age of the participants, however the standard deviation (SD) is missing. Moreover, the TERS score was presented as a range, mean, and median with SD, also the skewness to the right is elaborated…In my opinion, expressing data as mean ± SD is not entirely justified. Additionally, Table 1 (Sociodemographic characteristics of the sample at T0) would be more informative for the reader if it included numbers along with percentages, rather than just percentages.
The Discussion is elaborated and founded on presented results. The strengths and weaknesses of the study are clearly presented. However, the fact that all included patients were Caucasians should be mentioned in the text earlier.
Moreover, more than half (29/48) of references were published before 2018.
To conclude, the article presents novel results that might have clinical significance in the field of transplantation medicine. However, certain modifications should be made before the final decision on whether to accept or not the article.
Author Response
Response to Reviewer Comments
Manuscript submitted to JCM: “Predictive Value of the Transplant Evaluation Rating Scale in Living-Donor Kidney Transplant Recipients: A Retrospective Analysis”
Dear Reviewer,
We sincerely thank you for your comprehensive and constructive review of our manuscript. Your thoughtful comments and suggestions have been instrumental in improving the clarity and robustness of our work. Below, we provide a detailed, point-by-point response to your feedback, outlining the changes made to the manuscript accordingly.
Study Design Clarification
Comment:
"The authors claim to present a prospective single-centre cohort study; however, the TERS rating was conducted retrospectively. The design of the study is not completely clear."
Response:
We acknowledge that the retrospective application of TERS may lead to ambiguity regarding the study design. To address this, we have clarified in the Methods section (Z198-205) that while data collection for patient outcomes was conducted prospectively, the TERS scoring was applied retrospectively based on pre-existing psychosocial evaluation reports. This approach was necessary as TERS was not routinely implemented in clinical practice during the study period. Similar retrospective applications have been successfully employed in other studies to leverage available data without imposing additional burdens on patients​​.
Missing Standard Deviation and Data Presentation
Comment:
"The mean age of participants was presented, but the standard deviation (SD) is missing. Additionally, Table 1 would be more informative if it included both numbers and percentages, rather than just percentages."
Response:
Thank you for pointing this out. We have now included the standard deviation (SD) of the participants’ mean age in Table 1 (Z342). Additionally, we have revised Table 1 to include both absolute numbers and percentages for all sociodemographic variables to improve clarity and readability for the reader (Z342).
Data Presentation of TERS Scores
Comment:
"The TERS score was presented as a range, mean, and median with SD, but presenting data as mean ± SD may not be entirely justified. Consider including interquartile range (IQR)."
Response:
We agree with your suggestion and have now included the interquartile range (IQR) alongside the mean and median TERS scores to provide a more comprehensive view of the data distribution (Z335).
Ethnicity of Participants
Comment:
"The fact that all included patients were Caucasians should be mentioned earlier in the text."
Response:
We have now added this information to the Results section to ensure transparency regarding the study population’s demographic characteristics (Z332). This addition provides context for the generalizability of our findings.
Age of References
Comment:
"More than half of the references were published before 2018."
Response:
We acknowledge that some references are older, including foundational works such as Twillman’s original 1993 TERS publication. This reflects the limited availability of recent research on TERS. Despite this, we have conducted a thorough literature search to include the most relevant and updated sources where available. We appreciate your understanding of the challenges in sourcing newer studies for this specific instrument.
Discussion and Conclusion
Comment:
"The Discussion is well-elaborated and founded on presented results. However, certain modifications should be made before the final decision."
Response:
We have further refined the Discussion to emphasize the study’s strengths and limitations, explicitly addressing the retrospective nature of TERS application and its implications. Additionally, we have reiterated the study’s novelty and clinical significance in the Conclusion, ensuring the manuscript aligns with your suggestions.
We hope these revisions address your concerns and improve the manuscript’s quality. We are grateful for your valuable input, which has helped us enhance the clarity and impact of our work.
Thank you again for your thoughtful review.
Sincerely,

Round 2
Reviewer 1 Report
Comments and Suggestions for Authors
Well done.